# Cytotaxonomic characterization and estimation of migration patterns of onchocerciasis vectors (*Simulium damnosum sensu lato*) in northwestern Ethiopia based on RADSeq data

**Shannon M. Hedtke**[1], **Rory J. Post**[2,3], **Sindew Mekasha Feleke**[4], **Fikre Seife Gebretsadik**[5], **Daniel A. Boakye**[6], **Andreas Krueger**[7], **Warwick N. Grant**[1], **Craig S. Wilding**[2]*

**1** Department of Environment and Genetics, School of Agriculture, Biomedicine and Environment, La Trobe University, Bundoora, Victoria, Australia, **2** School of Biological and Environmental Sciences, Liverpool John Moores University, Liverpool, United Kingdom, **3** Disease Control Department, London School of Hygiene and Tropical Medicine, London, United Kingdom, **4** Ethiopia Public Health Institute, Addis Ababa, Ethiopia, **5** Neglected Tropical Disease Prevention and Control Program, Federal Ministry of Health, Addis Ababa, Ethiopia, **6** Parasitology Department, Noguchi Memorial Institute for Medical Research, Accra, Ghana, **7** Military Hospital Hamburg, Department Tropical Medicine, Hamburg, Germany

* C.S.Wilding@ljmu.ac.uk

**Data Availability Statement:** All RADseq data are deposited in GenBank as a Sequence Read Archive

## Abstract

### Background

While much progress has been made in the control and elimination of onchocerciasis across Africa, the extent to which vector migration might confound progress towards elimination or result in re-establishment of endemism in areas where transmission has been eliminated remains unclear. In Northern Ethiopia, Metema and Metekel—two foci located near the Sudan border—exhibit continuing transmission. While progress towards elimination has been faster in Metema, there remains a problematic hotspot of transmission. Whether migration from Metekel contributes to this is currently unknown.

### Methodology/Principle findings

To assess the role of vector migration from Metekel into Metema, we present a population genomics study of 151 adult female vectors using 47,638 RADseq markers and mtDNA *CoI* sequencing. From additional cytotaxonomy data we identified a new cytoform in Metema, closely related to *S. damnosum s.str*, here called the Gondar form. RADseq data strongly indicate the existence of two distinctly differentiated clusters within *S. damnosum s.l.*: one genotypic cluster found only in Metema, and the second found predominantly in Metekel. Because blackflies from both clusters were found in sympatry (in all four collection sites in Metema), but hybrid genotypes were not detected, there may be reproductive barriers preventing interbreeding. The dominant genotype in Metema was not found in Metekel while the dominant genotype in Metekel was found in Metema, indicating that (at the time of sampling) migration is primarily unidirectional, with flies moving from Metekel to Metema. There

(Bioproject PRJNA819953). Unique CoI sequences have been submitted to NCBI's GenBank database (https://www.ncbi.nlm.nih.gov/) with accession numbers OM866887-OM866988.

**Funding:** The genetic analysis work in this study was financially supported by the Global Challenge Research Fund (to CSW) and field collections were funded by the End Fund inc. The funders had no role in study design, data collection and analysis, decision to publish, or preparation of the manuscript. No authors received any salary from the funder of this study.

**Competing interests:** The authors have declared that no competing interests exist.

was strong differentiation between clusters but little genetic differentiation within clusters, suggesting migration and gene flow of flies within the same genetic cluster are sufficient to prevent genetic divergence between sites.

## Conclusions/Significance

Our results confirm that Metekel and Metema represent different transmission foci, but also indicate a northward movement of vectors between foci that may have epidemiological importance, although its significance requires further study.

### Author summary

Onchocerciasis is a severely debilitating disease caused by infection with a parasitic nematode worm (*Onchocerca volvulus*) transmitted from person to person by blood-sucking blackflies (Diptera: Simuliidae). Ethiopia has set a goal to eliminate the transmission of onchocerciasis through administration of the drug ivermectin, with 80% coverage, to all endemic areas. One potential challenge that might impede elimination in Ethiopia (and elsewhere in Africa) is the movement of vector blackflies carrying parasites between endemic areas, leading to failure to eliminate transmission. Cytogenetic data are the primary means for delimiting vector species, and we report the detection of a new cytotaxonomic form in northwestern Ethiopia, the "Gondar form" of *Simulium damnosum s.l.* Comparisons of nuclear genetic diversity of vectors in northwest Ethiopia suggests genetic differentiation consistent with reduction in mating between flies from two foci of ongoing transmission–Metema and Metekel. These results further suggest movement of blackflies is likely from Metekel, an area with ongoing transmission, into Metema, an area where persistent hotspots of transmission have been recently identified. Further research is warranted to determine whether the migration of blackflies between these areas that results in the observed genetic similarity is occurring at a timescale epidemiologically relevant to the persistence of these transmission hotspots.

## Introduction

Onchocerciasis (river blindness) is a severely debilitating parasitic disease caused by infection with the filarial nematode *Onchocerca volvulus*. Symptoms include impaired vision (including blindness), skin disease (skin lesions, intolerable itching, and depigmentation leading to social exclusion), and occasional neurological-endocrine disorders (epilepsy and Nakalanga syndrome) [1–4]. Worldwide, approximately 244 million people are at risk of onchocerciasis, with >99% of affected individuals living in sub-Saharan Africa [5]. Annual, biannual, or even quarterly mass drug administration of ivermectin (MDA) was effective at eliminating transmission in most endemic areas in the Americas; MDA implemented using community-directed treatment with ivermectin (CDTI), initially by the African Programme for Onchocerciasis Control (APOC; 1995–2015), has been similarly effective at eliminating transmission in some parts of Africa [6,7]. Consequently, endemic countries in Africa with encouragement, co-ordination, and advice from the World Health Organization (WHO) have declared their intention to eliminate transmission [6], and in a significant number of endemic areas it has been possible to stop treatment because transmission has already been interrupted.

However, these early successes tended to have been in areas where onchocerciasis was endemic in isolated foci (of various sizes) such as those in Uganda, Sudan, and countries in the Americas [8]. Infective vector migration might interfere with progress towards elimination where endemic areas are geographically closer together or continuously distributed, as is the case for much of sub-Saharan Africa [9]. Long-distance migration by infective adult female blackflies of some vector species in West Africa brought significant numbers of parasites into areas where they had been controlled by larviciding during the WHO Onchocerciasis Control Programme (1975–2022) (before the widespread adoption of CDTI) and were considered a serious threat to the control of onchocerciasis [10–12]. Migration by infective vectors could similarly contribute to failure to achieve elimination of transmission by CDTI, or might result in re-establishment of endemism in areas where onchocerciasis was previously considered to have been eliminated ([13–15]; see also e.g., [16–18] with regards to mosquito vectors).

In Ethiopia, over 17 million people live in endemic areas or are at risk of infection [19]. In 2013, the Ethiopian Federal Ministry of Health launched the Ethiopian Onchocerciasis Elimination Program with the initial goal of interrupting transmission by 2020 [19]. The main control strategy is CDTI delivered twice per year, with a goal of maintaining treatment coverage at >80% across all known endemic districts. While there has been considerable success, transmission has not been eliminated throughout the country and onchocerciasis remains an issue in some regions. In the Amhara region of Ethiopia, approximately 2 million people are at risk of onchocerciasis, including in the Ethiopian part of the Metema-Galabat cross-border focus (which consists of the Metema sub-focus in Ethiopia and the Galabat sub-focus directly adjacent in Sudan). In the Metema-Galabat focus, onchocerciasis has been well controlled, although a transmission hotspot has been identified on the Wudi Gemzu River in Metema [20]. Transmission of the parasite also continues in the Metekel focus 20 km to the south [20]. While CDTI has been increased to three times per year in the Metema hotspot [20], one concern is that blackfly immigration from Metekel could be responsible for ongoing transmission in the hotspot and/or re-introduce transmission of onchocerciasis into Metema should CDTI be stopped. Whether adult flies from Metekel are migrating into the cleared areas of the Metema focus is unknown, but the distance between the two foci is certainly within the known distance for long-range wind-assisted migration [10,12,21,22]. The epidemiological risk of blackfly migration should be incorporated into decisions about whether to stop CDTI in these areas, but there are currently no routinely employed methods by which the pattern and extent of vector migration can be measured. Mapping and quantifying the movement of blackflies would be useful for determining optimal strategies for successful elimination.

The situation is considerably complicated by the multiplicity of vectors. In Ethiopia a number of *Simulium* species have been recorded as anthropophilic (S1 Table) and amongst these *Simulium damnosum s.l.*, the vector of most onchocerciasis in Africa (including Ethiopia), is not a single species but a species complex. The historical development of our understanding of the *S. damnosum* complex in Ethiopia has been previously described [23–26]. Studies of the polytene chromosomes found in the silk glands of larvae have revealed many structural mutations (mostly inversions) which, according to population cytogenetic analyses, indicate distinct species known as sibling species or cytospecies because they are morphologically more or less identical but reproductively isolated [27]. Because not all chromosomal rearrangements have been proven to lead to reproductively isolated species [27], we will use the more general term "cytoform" for blackflies that are structurally distinct. However, cytoforms commonly show important differences in their roles as vectors [28,29]. There are approximately 31 distinct cytoforms recognized within the *S. damnosum* complex across Africa, with four in Ethiopia: Kisiwani, *S. kaffaense*, *S. soderense*, and *S. kulfoense* [30]. Of these, cytoform *S. kaffaense* is currently the only proven vector in Ethiopia [23,24]. However, cytoforms have not yet been

identified in many onchocerciasis-endemic areas in Ethiopia, and it is likely that additional variation exists, especially in areas with distinctive ecology (such as Metema). One challenge is that blackflies only have suitable polytene chromosomes for identification in the silk glands of the larvae (homologous to the salivary glands of other dipteran larvae), which does not allow for the direct identification of biting adult females. Morphological or molecular traits associated with cytoform identity would be more convenient, but despite more than 40 years of research, these have proved generally difficult to find [28].

Population genetics may assist in the assessment of the role played by vector migration (if any) in ongoing transmission, because genetic sequence similarities and differences can be used to estimate the per-generation rate of movement of blackflies from one onchocerciasis endemic area to another. *Simulium damnosum s.l.* breeding sites are found in fast-moving waters and can be permanent, seasonal, or ephemeral in location and productivity. This demographic instability, combined with local breeding and short generation times, is expected to increase similarity within a transmission zone due to inbreeding, while also increasing differences from other locations due to stochastic changes in allele frequencies (i.e., genetic drift). This increases our ability to detect clusters of genetic diversity that may be relevant to how flies are moving across the geographic landscape, and to detect individual flies that are migrants. Quantification of the genetic similarities and differences in samples of blackflies that are competent vectors of *O. volvulus* may be used to determine whether ongoing or renewed transmission of the parasite is likely to be driven by movement of blackflies.

In theory, cytotaxonomic variation (in the form of inversion frequencies) may also give some qualitative indication of migration, because differences in inversion frequencies between localities indicate that blackflies from these localities are not mixing freely. However, in practice, the number of characters (inversions) available for analysis is too small for quantitative assessment, and the number of specimens from which cytological data is derived are low. Furthermore, because inversions cause physical linkage between genes, natural selection could disturb signals of migration by rapidly removing chromosomal variation. The use of genetic information less likely to be subject to selection would be more informative about gene flow between areas. DNA barcoding utilizing sequencing of the mitochondrial cytochrome oxidase 1 (*CoI*) gene [31] has been applied widely for studies of population subdivision, particularly species delimitation, including in blackflies (e.g. [32,33]). However, phylogenetic analyses of blackflies suggest that nuclear sequence data tend to be more congruent with species delimitation based on morphology and cytotaxonomy, while the mitochondrial genome appears prone to introgression, leading to stronger geographic similarities across species boundaries [34, 35]. Despite its epidemiological importance, there remains no nuclear genome available for any member of the *S. damnosum* species complex. However, recently developed reduced-representation protocols such as restriction-site associated DNA sequencing (RADseq) [36] can be used to interrogate variation at thousands of loci without the need for pre-existing genome assemblies.

Here, we describe analyses of chromosomal, mitochondrial, and nuclear genetic variation in anthropophilic blackflies from the Metema and Metekel regions of Ethiopia. The objectives were to identify vector cytoforms in Metema, and to examine patterns of genetic variation to explore the potential role of vector movement in transmission of *O. volvulus* within and between the onchocerciasis-endemic areas of Metema and Metekel.

## Materials and methods

### Study area

The study area consisted of a number of sub-parallel river basins running northwesterly from the Ethiopian highlands to join the River Nile. The Metema-Galabat onchocerciasis focus is

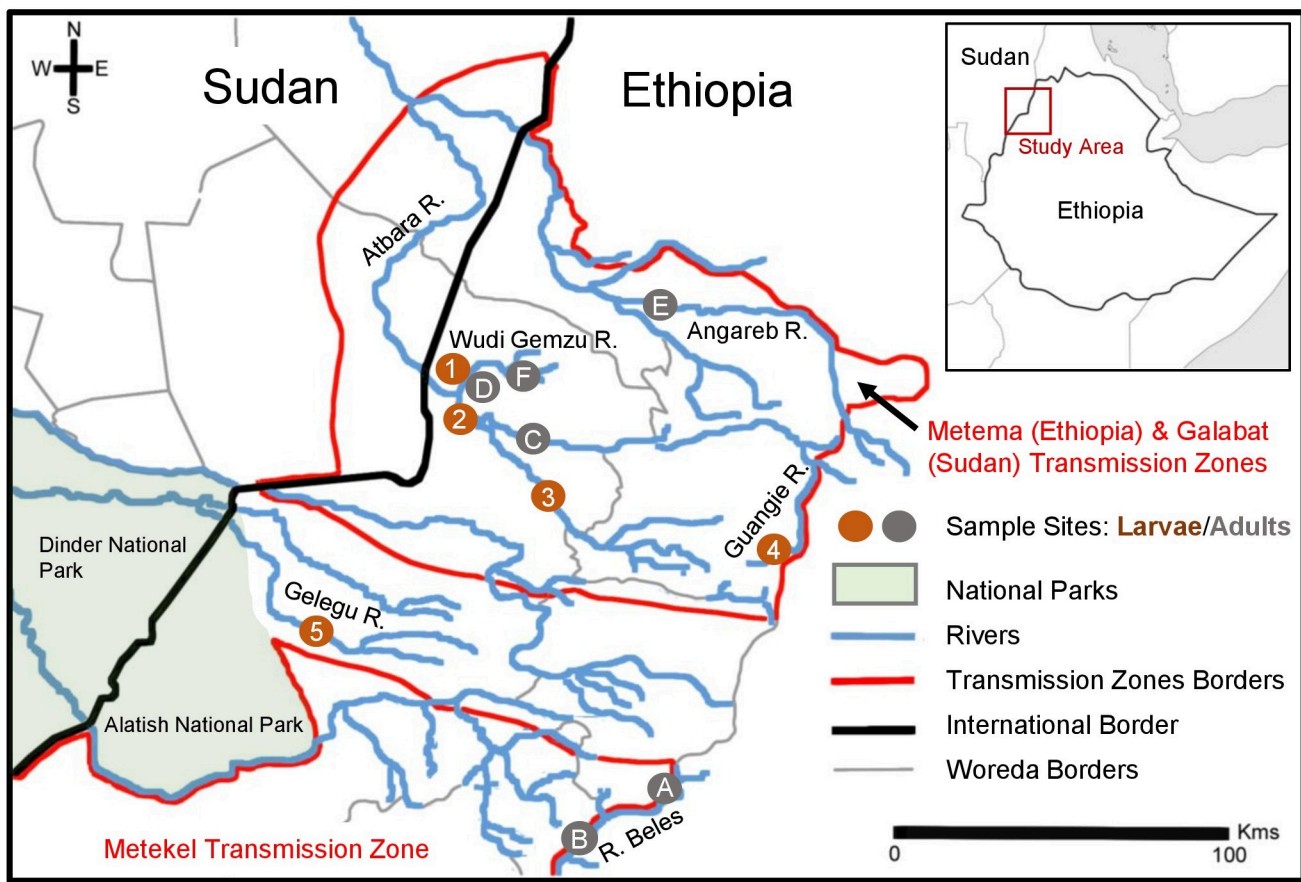

**Fig 1. Map showing the Metema-Galabat and Metekel foci of onchocerciasis and collection sites sampled for *Simulium* in northwest Ethiopia.** Transmission zone borders are outlined in red. Numbers/red circles (1–5) indicate larval samples collected in 2013 for cytotaxonomy and letters/grey cicles indicate blood-seeking adult females collected in 2017–8 for DNA sequencing (see Tables 1 and S2). Basemap property of the Government of Ethiopia and supplied by the Ethiopian Ministry of Health.

centered on the Atbara river basin in Ethiopia and Sudan, but in Ethiopia it also includes some parts of the Angereb river basin which lies to the north of the Atbara. The two rivers flow NNW separately out of Ethiopia to join each other in Sudan and eventually to join the River Nile at Atbara town (north of Khartoum). South of the Metema sub-focus, an area free of onchocerciasis occupies the Shinfa river basin which flows NW into Sudan and joins the Blue Nile at Wad Madani. This area is referred to as a 'buffer zone' because it is situated between the Metema sub-focus to the north and the Metekel focus to the south in the Dinder River basin, which also flows NW into Sudan joining the Blue Nile upstream of Wad Madani (separately from the Shinfa). To the south of the Dinder river basin is the Blue Nile itself (Fig 1).

The altitude of the study area rises from 500 m near the border with Sudan to approximately 2000 m. The geology is largely Mesozoic volcanic basalts with many faults [37], and is thus suitable for the development of broken-water rapids and vector breeding sites. Meteorological data (www.ethiomet.gov.et –accessed 2013) indicates that in NW Ethiopia there is a single dry season October-April and a single rainy season May-September. The prevailing wind comes from the northeast during the dry season and from the south or southwest during the rainy season [38,39]. River discharge is undoubtedly seasonal, but there is little evidence about the extent to which rivers and streams actually cease to flow. Total rainfall is 800–1200 mm

**Table 1. *Simulium* collection sites in northwestern Ethiopia.** Larvae collected in 2013 (sites 1–5) were used for cytotaxonomy, and adult females (f) collected in 2017–18 (sites A-F) were used for DNA sequencing. See also Fig 1 and S2 Table.

| Site | Transmission zone | Locality | River | Life stage | Number +*S. bovis* | Year of collection |
|---|---|---|---|---|---|---|
| 1 | Metema | Wudi Gemzu | Wudi Gemzu | larvae | 10 | 2013 |
| 2 | Metema | Guange N˚ 6 | Kibe | larvae | 3 | 2013 |
| 3 | Metema | Meka | Meka | larvae | 2 | 2013 |
| 4 | Metema | Dubaba Kebele | Guangie | larvae | 17 | 2013 |
| 5 | Metema | Delegu Anana | Delegu Namo | larvae | 4 | 2013 |
| A | Metekel | Selassie Godiguadit | Tana Beles | adult f | 33+0 | 2017 |
| B | Metekel | Block 4 | Abat Beles | adult f | 33+0 | 2017 |
| C | Metema | Asakefari | Asakefari | adult f | 15+16 | 2018 |
| D | Metema | Wudi Gemzu | Wudi Gemzu | adult f | 29+2 | 2018 |
| E | Metema | Kisha | Angareb | adult f | 6+23 | 2017–18 |
| F | Metema | Nega Wuha | Wudi Gemzu | adult f | 32+0 | 2018 |

rainfall p.a., and the natural vegetation is a form of open savanna woodland known as 'Combretum-Terminalia woodland and wooded grassland' [40].

## Cytotaxonomy

Larval specimens of the *S. damnosum* complex were collected in October 2013 (by DAB) from the Metema sub-focus of the Metema-Galabat onchocerciasis focus and from the buffer zone south of the Metema sub-focus, all in the North Gondar Zone of Amhara Region (Fig 1 and Tables 1 and S2). Larvae were hand-picked using forceps from vegetative and stony substrates found in white-water rapids and preserved in Carnoy's fixative according to standard methods [41] and subsequently maintained at 4˚C (except during transport at ambient temperature). Larvae of *S. damnosum s.l.* were identified according to Davies and Crosskey [41], and chromosome preparations were made from larval silk glands using standard methods and compared with the banding sequences of published chromosome maps [24,42–44].

## Collection of anthropophilic adult female blackflies

Adult female appetitive blackflies were collected at various times during the period April 2017 to August 2018 from the Metema sub-focus (four sites) and the northern part of the Metekel focus (two sites), (Fig 1; Tables 1 and S2), using human landing catches between 07:00–18:00 by Ethiopia Ministry of Health and the Public Health Institute (EPHI) and staff using standard procedures [41]. Ethical approval was obtained from the Ethiopian Public Health Institute (EPHI) and local administrative offices prior to collection. All staff involved in collections received standard training, consented to undertake fly collections, and received ivermectin as part of the routine MDA. Flies were preserved in 95% ethanol in the field and subsequently maintained at 4˚C (except during transport at ambient temperature). Preliminary morphological identification of flies was undertaken by EPHI in Ethiopia to remove any non-blackflies, and specimens were identified to morpho-species by RJP and CSW (according to [41,45,46]). Two species were identified among catches: *S. damnosum s.l.* and *Simulium bovis*.

## DNA extraction from adult female blackflies

DNA was extracted from 192 specimens, including 41 *S. bovis* and 151 *S. damnosum s.l.*, using a GeneJet genomic DNA extraction kit (Thermo Fisher Scientific, Altrincham, UK) following the manufacturer's instructions. DNA concentration was assessed using the Qubit dsDNA HS Assay Kit (Invitrogen, Thermo Fisher Scientific).

## mtDNA barcoding

Partial mtDNA *CoI* fragments were amplified using the primers of Folmer *et al.* [47] at 0.2 μM using 1× GoTaq HS mastermix (Promega Corp., Madison, Wisconsin, USA) with cycling conditions of 95°C for 3 min followed by 35 cycles of 95°C for 1 min, 40°C for 1 min, and 72°C for 1.5 mins, with a final extension step at 72°C for seven minutes. PCR products were purified using a GeneJet PCR purification kit (Thermo Fisher) and sequenced by GATC Biotech, Konstanz, Germany. Sequences were aligned using *ClustalW* v 2.0 [48]. A haplotype network of all available haplotypes was constructed using the TCS algorithm in *PopART* v 1.7 [49] and variability and dN/dS statistics calculated in *DnaSP* v 6 [50].

## RADseq

A minimum of 100ng DNA was used for RADseq following digestion with *Msl*I. Libraries were sequenced on an Illumina NextSeq 500/550 by LGC Genomics GmbH (Berlin, Germany) using v2 chemistry with 150 bp paired-end reads. Reads were filtered for quality and adapter sequence and the *Msl*I restriction enzyme cut sites removed. Orthologs were identified and extracted using the program *Stacks* [51], with further filtering for quality [52] and parameter settings chosen using the r80 method [53] (S1 Text).

## Population statistics and differentiation

Basic population genetic diversity statistics for each sampling location were estimated using *Stacks* [51]. Principal components (PCs) of genetic variation were plotted using the R library package *adegenet* v. 2.1.3 [54,55]. Linkage disequilibrium was estimated using $r^2$ for unphased genotypes, and sites with an estimated $r^2 > 0.1$ were pruned such that only the first site was retained using *vcftools* v. 0.1.16 [56]. Two alternative algorithmic approaches for identifying subpopulations of blackflies and computing the posterior probability of each blackfly's membership in each inferred subpopulation were used: (1) discriminant analysis of principle components (DAPC; [57]) using *adegenet* and (2) the program *structure* v.2.3.4 for Linux [58].

   DAPC was used to infer the number of clusters or subpopulations suggested by the data using the Bayesian information criterion (BIC) and to assign individuals to those subpopulations. In a separate analysis, maximizing and plotting the differentiation among the six sampling locations was explored. *Structure* [58] was used to both define the number of subpopulations and to assign individuals to those sub-populations. For specifics on parameterization of these analyses, see S1 Text.

# Results

## Cytotaxonomy

Table 2 shows the inversions that are characteristic and diagnostic of the cytospecies and other cytoforms that are members of the *S. damnosum* subcomplex within the *S. damnosum* complex (i.e., they are closely related to *S. damnosum* s.str. [27]). Table 3 shows the status of inversions within the material examined from Metema. The frequencies of inversions in Metema, distribution between the sexes (potential sex-linkage), and photographs of selected chromosomal inversions are presented in S3–S6 Tables and S1–S6 Figs.

   Only 36 specimens were analysed from Metema, but it was clear that all specimens belonged to the *S. damnosum* subcomplex because of their fixed inversions, especially 2L-C. However, beyond that, they did not correspond precisely to any known cytoform. Specimens homozygous for 2L-C.8 would normally be identified as *Simulium sirbanum*, while 2L-8 is

**Table 2. Cytotaxonomic summary of the different members of the *Simulium damnosum* subcomplex (modified from [27]).**

| Cytoform | 1S | 1L | 2S | 2L | 3S | 3L |
|---|---|---|---|---|---|---|
| *S. damnosum* Nile form | **1**.2*.3* | **1.3** | St | **C** | St | **2** |
| *S. damnosum* Volta form | **1**.2.3 | **1.3** | St | **C**.8* | St | **2**.6☼ |
| *S. sirbanum* sirba form | **1.2**.3* | **1.3** | St | **C.8** | St | **2**.6♦ |
| *S. sirbanum* sudanense form | **1.2**.3* | **1.3** | St | **C.8** | St | **2**.6♦ |
| *S. sirbanum* type IV | **1.2.3** | **1.3** | St | **C.8** | St | **2**.(6?) |
| Hamadense | **1.2.3** | **1.3** | St | **C.8** | St | **2.6** |

Notes: Fixed inversions are bold and underlined

*Sex-linked inversions

♦Inversion normally at high frequency

☼Inversion normally at low frequency. St denotes the standard chromosome sequence arbitrarily defined as that shown by *Simulium kilibanum* [34].

expected to be absent or sex-linked polymorphic in *S. damnosum s.str*. In the samples from North Gondar, it was found to be polymorphic but not sex-linked.

There are no chromosomal data available from other populations of the *S. damnosum* subcomplex in Sudan, South Sudan, or Ethiopia, except for Abu Hamed on the River Nile in Sudan [59]. The population examined from North Gondar is genetically differentiated from the Hamedense cytoform by eight inversions. Hamedense has no polymorphic inversions and nine fixed inversions (including 2L-8). Four of these fixed inversions are polymorphic in North Gondar, Ethiopia, with five additional polymorphic inversions unique to North Gondar. The populations in North Gondar zone and Abu Hamed are clearly different from each other, but there could still be some low-level genetic exchange.

## Mitochondrial DNA barcoding

645 bp of *CoI* sequence was amplified successfully from 189 of the 192 samples. Unique sequences have been submitted to NCBI's GenBank database (https://www.ncbi.nlm.nih.gov/) with accession numbers OM866887-OM866988. The haplotype network shows three main groupings of sequences with *S. damnosum s.l.* distinct from *S. bovis* (Fig 2). However, in addition to the main body of *S. damnosum s.l.* specimens, a small number ($n = 6$) appear genetically distinct from the other *S. damnosum s.l.* specimens, being separated by a minimum of 27 mutational steps. Although the closest match through a *BLAST* analysis [60] is in the *S. damnosum* species complex from Benin (Accession KF839980.1 [61]), due to the differences in sequence these are subsequently referred to here as "query"-*Simulium damnosum*.

**Table 3. Cytotaxonomic summary of the *Simulium damnosum* subcomplex from the study area (Amhara Region, Ethiopia).**

| Site | Sample River | River Basin | *n* | 1S | 1L | 2S | 2L | 3S | 3L |
|---|---|---|---|---|---|---|---|---|---|
| 1 | Wudi Gemzu | Atbara-Angereb | 10 | **1**.2.3 | **1.3**.22*.23* | St | **C**.8.2b* | St | **2.6** |
| 2 | Kibe | Atbara-Angereb | 3 | **1**.2.3 | **1.3** | St | **C**.8.2b* | St | **2.6** |
| 3 | Meka | Atbara-Angereb | 2 | **1.2**.3 | **1.3** | St | **C.8** | St | **2.6** |
| 4 | Guangie | Blue Nile-Rahad | 17 | **1**.2.3 | **1.3**.22*.23*.24* | St | **C**.8.2b*.70* | St | **2.6** |
| 5 | Delegu Namo | Blue Nile-Dinder | 4 | **1.2.3** | **1.3** | St | **C.8** | St | **2.6** |

Notes: Inversions which were only found in the homozygous state and are probably fixed inversions are shown in bold and underlined (although samples from Meka and Delegu Namo rivers were very small, and it is quite likely that inversion 1S-3 was polymorphic in these populations, as it is in the other three rivers); *New inversions unknown outside Ethiopia.

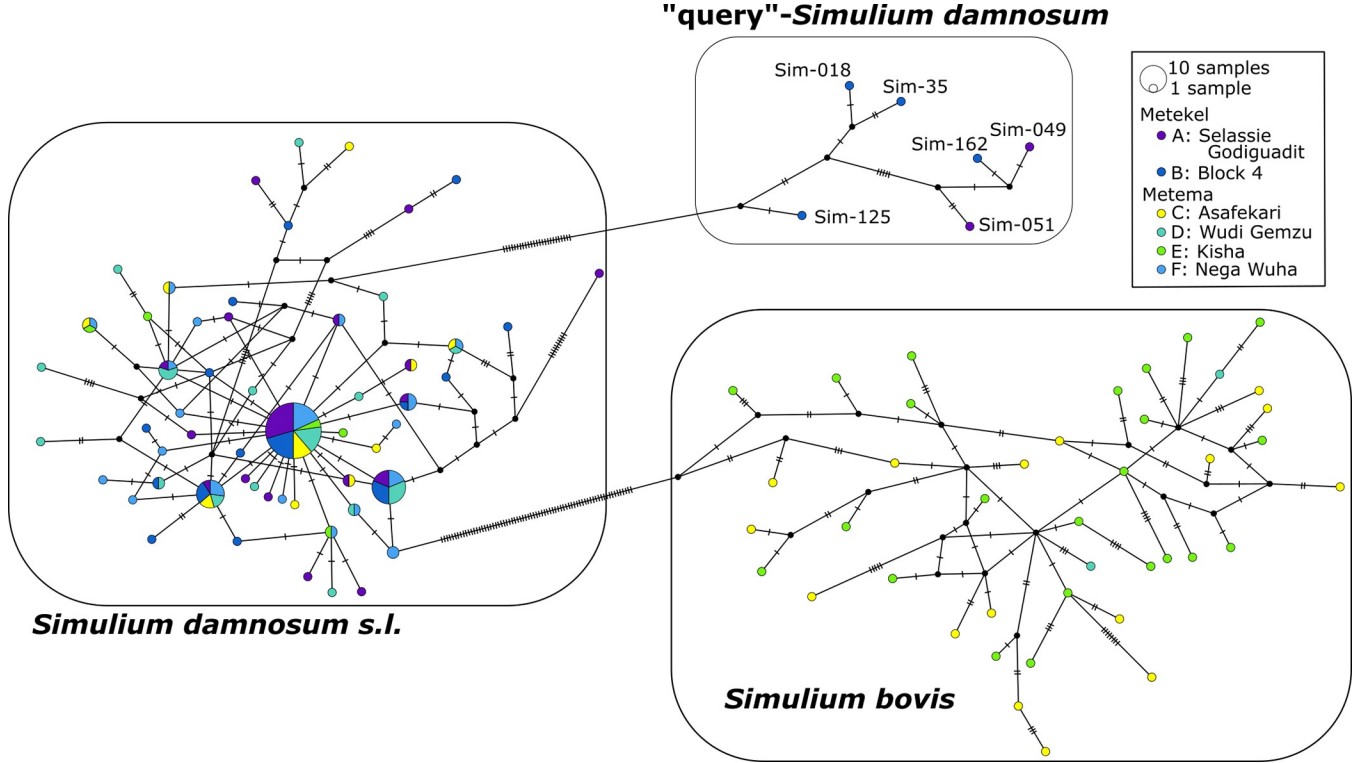

**Fig 2. Mitochondrial *CoI* haplotype network of 189 *Simulium* from Metema and Metekel foci.** Circles represent unique haplotypes; the size of the circles indicate the number of blackflies with that haplotype. Hatches along the lines connecting haplotypes indicate the number of sequence differences between them. Colors indicate the proportion of each haplotype found in each sampling location.

There were geographic differences in the distribution of *S. bovis*, *S. damnosum*, and "query"-*Simulium damnosum*. *Simulium bovis* was found only in 4 sites in Metema (C, D, E, F) and at sites D and F only at low frequency, while query-*damnosum* were found only at sites A and B in Metekel (Table 4).

**Table 4. Numbers of *Simulium damnosum* s.l., query-*S. damnosum* s.l., and *S. bovis* as determined by mitochondrial DNA *CoI* analysis among 192 specimens extracted for molecular analysis.** Numbers in parentheses represent samples morphologically identified as *S. damnosum* s.l. but which failed to amplify usable *CoI* barcodes.

| Collection Site (River) | | Species | | |
|---|---|---|---|---|
| | | *S. damnosum* s.l. | "Query" *S.damnosum* | *S.bovis* |
| A | Selassie Godiguadit (Tana Beles) | 31 | 2 | 0 |
| B | Block 4 (Abat Beles) | 29 | 4 | 0 |
| C | Asakefari (Asakefari) | 15(+2) | 0 | 16 |
| D | Wudi Gemzu (Wudi Gemzu) | 29(+1) | 0 | 2 |
| E | Kisha (Angareb) | 6 | 0 | 23 |
| F | Nega Wuha (Wudi Gemzu) | 32 | 0 | 0 |

Haplotype diversity based on *CoI* was high for *S. damnosum* ($\pi_h = 0.885$), and for "query"-*Simulium damnosum* and *S. bovis*, each haplotype was unique ($\pi_h = 1$; S7 Table). All *S. damnosum* s.l. populations (with the exception of site E, where there were only 6 samples sequenced) displayed significantly negative Tajima's D statistics (over all *S. damnosum* s.l. combined, Tajima's D = -2.236; S7 Table), indicative of the action of either a recent selective sweep of the mitochondrial lineage or a recent population expansion.

### RADseq Stacks parameter estimation

**All RADseq data are deposited in GenBank as a Sequence Read Archive (Bioproject PRJNA819953).** Here, we report RADSeq results only from anthropophilic *S. damnosum s.l.* samples (*N* = 151). Per-fly sequencing statistics (at M = 1) can be found in S8 Table. However, seventeen flies were removed from downstream analysis as their estimated average depth was lower than 10 when M = 1 or M = 8: two from Selassie Godiguadit (A: SIM-68, SIM-179), two from Block 4 (B: SIM-53, SIM-106), five from Asafekari, Metema (C: SIM-19, SIM-21, SIM-37, SIM-111, SIM-210), five from Wudi Gemzu, Metema (D: SIM-41, SIM-59, SIM-95, SIM-166, SIM-221), two from Nega Wuha, Metema (F: SIM-64, SIM-136), and one from Kisha (E: SIM-116). Thus, 134 *S. damnosum s.l.* remained for downstream analyses: 62 collected in the Metekel region (A: 31, B: 31) and 72 collected in the Metema focus (C: 12, D: 26, F: 29, E: 5).

**RADseq population statistics.** After filtering, 2,506,166 bp were sequenced across 14,469 orthologous loci, within which 47,638 single nucleotide sites were variable in the dataset. Estimates of genetic diversity were similar across all populations, although the number of polymorphic sites and number of private alleles varied with the number of samples in the population (S7 Table). After pruning for linkage disequilibrium, 23,860 single nucleotide sites were variable in the dataset.

Two approaches were used to explore how flies sampled clustered with regards to geography, and both were congruent despite their methodological differences. DAPC is a multivariate approach using PCA to maximize differences among groups while minimizing variance within groups [57]. The first PCs from PCA were plotted and show strong genetic differentiation between samples collected in Metekel and those collected in Metema (Fig 3A). The first ten PCs were used for the DAPC analysis using six groups as determined by cross-validation (S7 Fig). DAPC maximizes the differentiation observed in PCs and is consistent with the PCA plot, with most flies from Metema clustered together and independent of flies from Metekel (Fig 3B). However, nine flies collected from sampling locations within the Metema focus do not fall within this cluster, instead grouping with flies from Metekel (SIM-24, SIM-29, SIM-43, SIM-97, SIM-151, SIM-154, SIM-173, SIM-186, SIM-188). *When adegenet was utilized for model-based K inference, the BIC supported a model of two clusters* (S8 Fig). With two clusters, 90 PCs were determined to be optimal for DAPC using cross-validation (S9 Fig), and cluster assignment was consistent with the six-population DAPC with regards to identifying flies from Metema that were genetically more similar to flies from Metekel (Fig 3C). We do note that the six flies with strongly differentiated mtDNA haplotypes (in the group 'query-*S. damnosum*') are not differentiated from other flies on the basis of these nuclear markers (Figs 3A–3C).

The Bayesian model-based method implemented in the program *structure* estimates the cluster membership of each individual given a certain number of possible clusters, *K* [58]. The algorithm optimizes the groupings of those individuals to meet the assumptions of Hardy-Weinberg and linkage equilibrium. The results of population assignment did not change across all values of *K* tested (2 through 8) and were identical to the results from the DAPC analysis: nine flies collected in the Metema focus were inferred to be in the cluster which contained the flies from Metekel (Fig 3C).

## Discussion

The data presented here are from two distinct studies carried out to examine the cyto- and molecular genetic characteristics of vectors in the Metema and Metekel foci of northwestern Ethiopia. The cytogenetic study, carried out in 2013, sampled a relatively small number of larval *S. damnosum s.l.* and aimed to place vectors in the Metema focus into the current

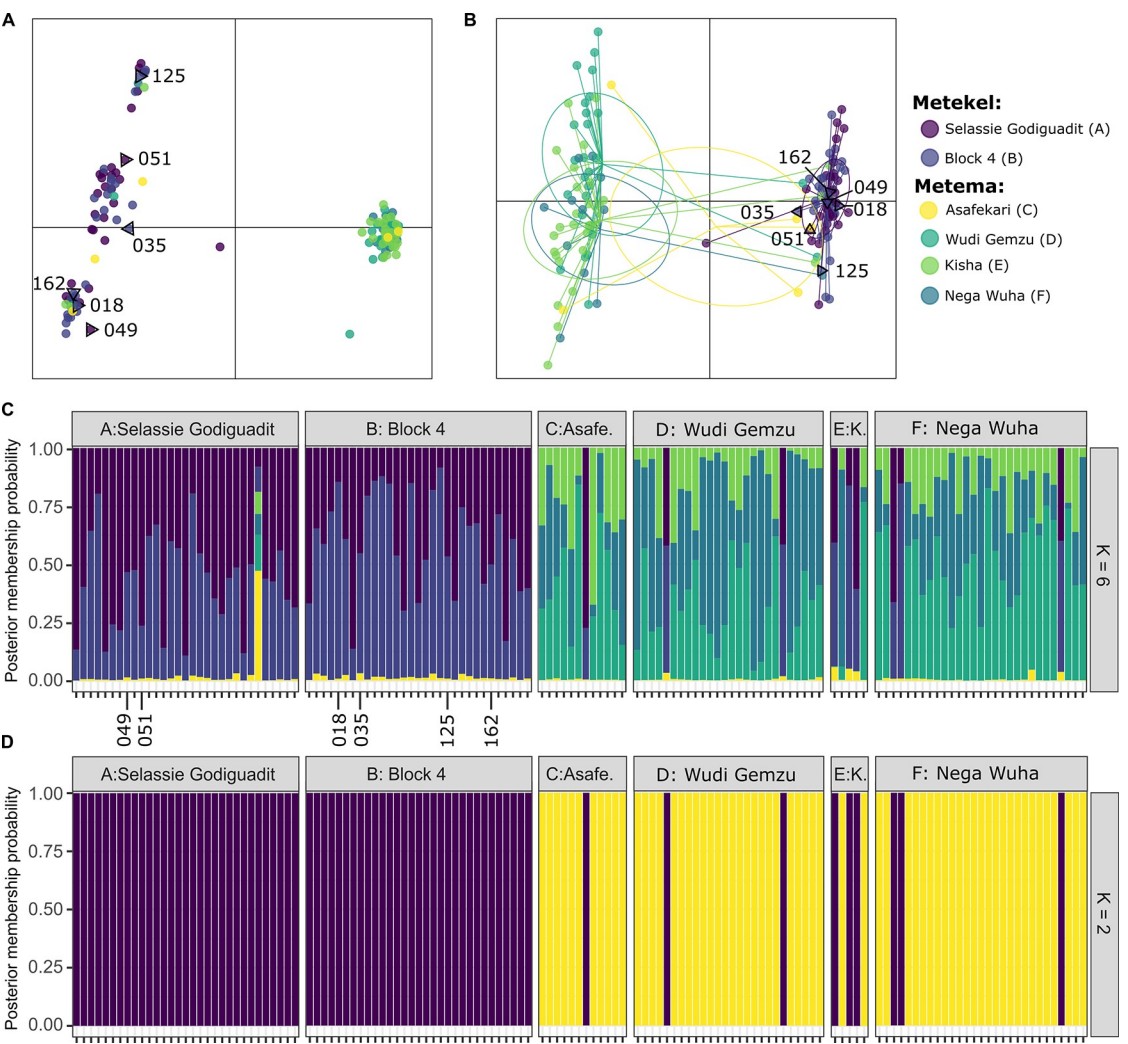

**Fig 3. Population genetic structure of *Simulium damnosum* from Ethiopia genotyped at 23,860 variant sites.** A) First (7.17% of total variance) and second (1.25% of variance) principal components (PCs) of variation in the genetic data (third and fourth PCs are presented in S10 Fig) and B) discriminant analysis of principal components (DAPC): each dot represents a genotyped fly; its color represents the sampling location. Lines connect to the center of the inertia ellipse for each sampling location. C) Genotype assignment of 134 flies based on DAPC for 6 sampling locations (K = 6) and D) based on assignment of each fly to one of two clusters (purple or yellow; K = 2). The results of *structure* with two to eight populations were identical to (D): all flies from Metekel were assigned to a cluster containing only Metekel flies, but 9 flies collected in Metema were inferred to be from this Metekel cluster. Numbered samples on A-C are the six "query"-*Simulium damnosum* specimens identified through mtDNA barcoding.

cytotaxonomic classification of *S. damnosum s.l.* DNA sequencing was performed using a larger, more geographically extensive sample of adult blackflies collected in 2018, and aimed to investigate the population structure of anthropophilic *Simulium* across the Metema and Metekel foci (without reference to their cytotaxonomic classification).

## Cytotaxonomy and the Gondar form of the S. damnosum complex

The specimens examined cytotaxonomically from North Gondar are a previously unknown, genetically distinct cytoform within the *S. damnosum* subcomplex, and we have named this cytoform the 'Gondar form'. It is unknown whether the Gondar form of the *S. damnosum* complex has a wider distribution in Ethiopia, or whether it extends into Sudan, because our

larval specimens were collected in 2013 and from the Metema sub-focus and the buffer zone immediately to the south only.

While we did not gather cytological data from samples collected in 2018, at least some of the adults collected in Metema may be the Gondar cytoform. Since no larvae were collected in Metekel, the adult blackflies collected there might be an as yet unidentified cytoform or could be a distinct lineage of the Gondar cytoform. Because rivers in northwestern Ethiopia are seasonal, breeding populations of *S. damnosum s.l.* are unstable, with vector migration between sites. It is therefore also possible that between 2013, when the blackflies used for cytotaxonomy were collected, and 2018, when collections for sequencing were done, the Gondar cytoform could have been replaced by some other cytoform(s) in the Metema sub-focus (e.g., fluctuations in cytoforms in Ghana [62]). This highlights one of the challenges in *Simulium* taxonomy as a whole: cytological data must be collected from larvae, whose adult characteristics may be unknown, while DNA is typically sequenced from adult flies, often collected because they are anthropophilic and thus potential disease vectors.

## Two Groups Revealed by RADseq Analyses of Genomic DNA

Analysis of DNA sequence data (RADseq and mtDNA) was carried out on adult blackflies collected in 2018 and found two genetically distinct clusters of *S. damnosum s.l.*: one from Metema sub-focus and one from Metekel focus, with no evidence of hybridization between them (Fig 3C and 3D). Blackflies from both clusters are found in the Metema focus sampling sites, while members of only one cluster were found across Metekel focus sites. Within both the Metekel and Metema foci, there was little genetic differentiation among blackflies, suggesting that interbreeding of blackflies within the same genetic cluster is sufficient to prevent population structure among sampling locations within each focus. Furthermore, between locations, there was no evidence of differentiation between the blackflies of the same cluster collected in both Metema and Metekel. This suggests that there is or has been gene flow between the two foci, and that this gene flow is likely to be unidirectional from Metekel to Metema. There are two possible explanations for this geneflow: (1) all sampled blackflies are the same species, and those from the second group collected in Metema are very recent migrants from Metekel that have not yet interbred with the majority Metema genetic cluster (and/or sample sizes were too low to detect hybrids) or (2) both clusters are endemic to Metema but they represent species that do not successfully interbreed, and migration from Metekel to Metema homogenizes genetic variation in that species found in both locations. While we do not have cytogenetic data from blackflies from either genetic cluster, reproductive isolation has been shown to evolve in other dipterans as a result of chromosome inversions, and inversions have contributed to adaptive variation for traits such as preference for indoors versus outdoors, optimal climate, or insecticide resistance in mosquitos (e.g., [63–65]).

Gene flow could occur as a result of either of two sorts of vector movements: long-distance migration (wind-assisted >100 km) and/or appetitive dispersal (active flight looking for a bloodmeal up to 20 km). The wind direction in this area is seasonal, from the southwest in the rainy season and from the northeast in the dry season. While northerly and southerly wind-assisted migration could both theoretically occur, ephemeral breeding sites affected by reduced rainfall would mean that far fewer flies would be available to be transported in the dry season than in the rainy season, when flies are abundant and winds are from the southwest. Such northerly migration could therefore result in long-distance migration from Metekel to Metema.

Appetitive dispersal does not occur over distances that would permit an individual blackfly to migrate through the buffer zone from Metekel to Metema. However, if suitable habitat is

continuous, regular small-scale migration could homogenize the blackflies in the two locations by gradual movement of genetic diversity across the landscape (i.e., gene flow through dispersal). This "stepping-stone" migration pattern would homogenize genetic diversity within a species, as might be expected to occur if the Metekel genetic cluster were a separate species with low differentiation across its sampled range.

The consequences of these two migration hypotheses on the epidemiology of onchocerciasis in northwestern Ethiopia are drastically different. If wind-dispersed in a single step, infective blackflies from an endemic area with ongoing transmission (Metekel) could cause hotspots of transmission to arise in an area close to elimination (Metema-Galabat) despite continuing drug treatment in Metema. If dispersal through the buffer zone is occurring in small steps between breeding sites over several generations, then the risk of recrudescence due to infective blackflies is low and would not need to be considered for stop-treatment decisions. Sampling within the buffer zone and/or sampling of blackflies at multiple time points would be required to discriminate between these hypotheses.

## Incongruent results from mitochondrial, nuclear, and cytotaxonomic data

While nuclear RADSeq markers effectively discriminate samples in our study, this is not the case for mitochondrial DNA. *CoI* haplotypes are shared across foci and provide no evidence of the discrimination of foci seen from multi-locus RADSeq markers. However, six flies from Metekel (two from Selassie Godiguadit [A] and four from Block 4 [B]) were genetically distinct based on mitochondrial genotyping and are clearly separated from the main body of *S. damnosum s.l.* samples in the mtDNA haplotype network (Fig 2). The high genetic distance between these six samples and all other *S. damnosum* is at odds with the nuclear DNA data (in which these six samples are indistinguishable from other Metekel flies). Previous phylogenetic analyses of blackflies from Africa have also found incongruence between trees built from mitochondrial data and those from both nuclear sequence data and cytotaxonomy (e.g., [34,35]). When incongruence is observed, it can be attributed to biological processes—such as incomplete lineage sorting (when shared ancestral polymorphism is randomly lost in one population) or introgression (interspecific hybridization followed by unidirectional backcrossing)—or to systematic errors in phylogenetic reconstruction [66], or even to PCR/sequencing error. Because of the substantial number of (presumably) independent nuclear loci reported here, and the depth of differentiation between the two mitotypes, systematic error is not a convincing explanation for the incongruence. However, testing the hypothesis for mitochondrial introgression from an unsampled species is not possible because of the absence of available *CoI* barcodes for all *Simulium* species in this region of Africa.

## Conclusions and significance

This study revealed that there was at least one new and unique cytotaxonomic form found within the *S. damnosum* complex in the study area and we have proposed to call it the Gondar form of the *S. damnosum* complex. The RADseq data indicated two genetically distinct clusters within the *S. damnosum* complex in the study area. It is possible, but unproven, that one or both of these genetic clusters is the novel cytotaxonomic form. Since these blackflies have not been characterized before, very little is known about whether they might exhibit traits which differ significantly from other known cytospecies for the transmission of onchocerciasis.

The RADseq analyses supports the hypothesis of northward geneflow from Metekel into Metema, but it remains unclear whether this represents long-distance movement of infective blackflies, which could cause a problem to onchocerciasis elimination efforts by bringing

parasites from Metekel into Metema, or gradual movement of genes north through the buffer zone, which would be less likely to introduce parasites. To quantify the risk of migration, it would be necessary to collect specimens regularly throughout the year and to demonstrate that at least some migrants were infective. However, this study has provided a proof of principal that nuclear genetic data (in this case partial sequencing using the RADseq approach) can reveal migration patterns in blackfly vectors of onchocerciasis.

## Supporting information

**S1 Text. Preparation and analysis of RADseq data.**
(DOCX)

**S1 Table. List of *Simulium* species recorded as anthropophilic in Ethiopia.**
(DOCX)

**S2 Table. Sample sites in northwestern Ethiopia where larvae or biting adult female *Simulium* sp. were collected.**
(DOCX)

**S3 Table. Inversion frequencies in the *Simulium damnosum* subcomplex from Ethiopia.**
(DOCX)

**S4 Table. Karyotype distribution of *Simulium damnosum* subcomplex from Ethiopia: Chromosome 2.**
(DOCX)

**S5 Table. Karyotype distribution of *S. damnosum* subcomplex from Ethiopia: Chromosome 1L.**
(DOCX)

**S6 Table. Karyotype distribution of *S. damnosum* subcomplex from Ethiopia: Chromosome 1S.**
(DOCX)

**S7 Table. Genetic diversity statistics across species and populations of *Simulium damnosum s.l.*, *S. bovis*, and the query-*damnosum* samples based on CoI DNA sequence data.**
(DOCX)

**S8 Table. Sequencing statistics for RADSeq data based on STACKS [1] when M = 1.**
(DOCX)

**S9 Table. Estimates of population genetic diversity of *Simulium damnosum sensu lato* samples collected from six locations in Ethiopia averaged across 47,638 variant sites.**
(DOCX)

**S10 Table. Estimates of pairwise genetic distance (below the diagonal) and $F_{ST}$ (above the diagonal) for six locations in Ethiopia based on 23,860 variant sites of *Simulium damnosum s.l.* collected in 6 locations in Ethiopia.**
(DOCX)

**S1 Fig. Inversion 2L-2b marked on chromosome homozygous 2L-C.2b/C.2b.**
(DOCX)

**S2 Fig. Heterozygous inversion 2L-70.**
(DOCX)

**S3 Fig. Homozygote 2L-C.8.70/C.8.70 showing breakpoints of inversion 2L-70.**
(DOCX)

**S4 Fig. Breakpoints of Inversions 1L-22, 1L-23 & 1L-24 on chromosome 1L-1.3/1.3. Also showing heterozygous inversion 1S-2 on chromosome 1S-2.3/3.**
(DOCX)

**S5 Fig. Heterozygous inversions 1L-22 & 1L-23 on chromosome 1L-1.3/1.3.22.23.**
(DOCX)

**S6 Fig. Heterozygous inversions 1L-23 & 1L-24 on chromosome 1L-1.3/1.3.23.24.**
(DOCX)

**S7 Fig. Cross-validation for determining the optimal number of principle components to use in a discriminant analysis of principle components when K = 6, the number of sampling sites for *Simulium damnosum s.l.* flies collected for sequencing.**
(DOCX)

**S8 Fig. Bayesian Information Criterion (BIC) for the number of clusters in nuclear sequence data of *Simulium damnosum s.l.* from Ethiopia from K = 1 through K = 40.**
(DOCX)

**S9 Fig. Results of cross-validation using 100 replicates for determining the optimal number of principle components to use in a discriminant analysis of principle components when K = 2, the number of clusters inferred based on nuclear sequence data of *Simulium damnosum s.l.* flies collected in Ethiopia.**
(DOCX)

**S10 Fig. The third (0.89% of all variance) and fourth (0.87%) principal components from analysis of 23,860 genetic variants in linkage equilibrium of *Simulium damnosum s.l.* blackflies collected at 6 sites in Ethiopia.**
(DOCX)

## Acknowledgments

We are grateful to Mr. Kalkidan Mekete, Mr. Gemechu Tadesse, Mr. Mulugeta Ejigu and Mr Kadu Meribo from EPHI and FMoH for conducting the field collections. We additionally thank Himal Shrestha and Emily Hendrickson at La Trobe University for useful discussion of the results and the WHO/APOC and The Carter Center staff who participated in supervision and training.

## Author Contributions

**Conceptualization:** Rory J. Post, Craig S. Wilding.

**Data curation:** Shannon M. Hedtke, Rory J. Post, Daniel A. Boakye, Andreas Krueger, Craig S. Wilding.

**Formal analysis:** Shannon M. Hedtke, Rory J. Post, Daniel A. Boakye, Andreas Krueger, Craig S. Wilding.

**Funding acquisition:** Craig S. Wilding.

**Investigation:** Shannon M. Hedtke, Rory J. Post, Sindew Mekasha Feleke, Fikre Seife Gebretsadik, Daniel A. Boakye, Andreas Krueger, Craig S. Wilding.

**Methodology:** Shannon M. Hedtke, Rory J. Post, Craig S. Wilding.

**Project administration:** Rory J. Post, Warwick N. Grant, Craig S. Wilding.

**Resources:** Rory J. Post, Sindew Mekasha Feleke, Fikre Seife Gebretsadik, Daniel A. Boakye, Andreas Krueger.

**Supervision:** Rory J. Post, Warwick N. Grant.

**Validation:** Shannon M. Hedtke, Rory J. Post, Craig S. Wilding.

**Visualization:** Shannon M. Hedtke, Rory J. Post, Sindew Mekasha Feleke, Fikre Seife Gebretsadik, Warwick N. Grant, Craig S. Wilding.

**Writing – original draft:** Shannon M. Hedtke, Rory J. Post, Craig S. Wilding.

**Writing – review & editing:** Shannon M. Hedtke, Rory J. Post, Warwick N. Grant, Craig S. Wilding.

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
