## [Decision Letter · Decision Letter 0]

18 Sep 2023

Dear Dr. Wilding,

Thank you very much for submitting your manuscript "Cytotaxonomic characterization and estimation of migration patterns of onchocerciasis vectors (Simulium damnosum sensu lato) in northwestern Ethiopia based on RADSeq data" for consideration at PLOS Neglected Tropical Diseases. As with all papers reviewed by the journal, your manuscript was reviewed by members of the editorial board and by several independent reviewers. The reviewers appreciated the attention to an important topic. Based on the reviews, we are likely to accept this manuscript for publication, providing that you modify the manuscript according to the review recommendations. 

Firstly, apologies for the delay in returning this manuscript - its was very challenging to find reviewers for this manuscript. I'm delighted that three very well-qualified reviewers all agree on the importance and high quality of the manuscript, and it will make a very nice contribution to PLoS NTD. Reviewers #1 and #3 have very few comments, athough I note the important point made by #3 that something should be said around the ethical and regulatory framework under which human landing catches took place.

The second reviewer makes some very helpful and detailed feedback to improve the manuscript. Most of these are small comments on the text - and I add some tiny edits I noticed in addition to these below. The most substantial suggestions that reviewer #2 makes are suggesting some more explicitly 'spatial' genetic approaches that might add to the presentation here. These are nice ideas, and I invite the authors to at least address this in a revision, but I leave it to their expertise whether they actually attempt these analyses or if they wish to merely address in the discussion what could be done in future: in particular, I suspect that the largely clustered adult sampling sites within each focus means that these spatially explicit approaches might not add enormously to the results already presented here.

Tiny editorial comments:

line 373: (in this case, Hardy-Weinberg equilibrium based on allele frequencies). I don't think this is quite right, as 'Hardy-Weinberg equilibrium' isn't a parameter of the structure model - its an assumption of the method that subpopulations are in HWE.

Line 437: from southwest in rainy season and from northeast in dry season -> "from the southwest in the rainy season and from the northeast in the dry season"

line 438 - the section on seasonal wind direction was just a tiny bit hard to follow - the message is clear once you've read it twice: that in the dry season few flies are around, so the prevailing wind from Metema -> Metekel is unlikely to lead to much migration, while the prevailing wind direction in the rainy season will lead to movement Metekel -> Metema. But I wonder if the authors can revise this to make this message a bit more explicit -

Sincerely,

James Cotton

Academic Editor

Nigel Beebe

Section Editor

Firstly, apologies for the delay in returning this manuscript - its was very challenging to find reviewers for this manuscript. I'm delighted that three very well-qualified reviewers all agree on the importance and high quality of the manuscript, and it will make a very nice contribution to PLoS NTD. Reviewers #1 and #3 have very few comments, athough I note the important point made by #3 that something should be said around the ethical and regulatory framework under which human landing catches took place.

The second reviewer makes some very helpful and detailed feedback to improve the manuscript. Most of these are small comments on the text - and I add some tiny edits I noticed in addition to these below. The most substantial suggestions that reviewer #2 makes are suggesting some more explicitly 'spatial' genetic approaches that might add to the presentation here. These are nice ideas, and I invite the authors to at least address this in a revision, but I leave it to their expertise whether they actually attempt these analyses or if they wish to merely address in the discussion what could be done in future: in particular, I suspect that the largely clustered adult sampling sites within each focus means that these spatially explicit approaches might not add enormously to the results already presented here.

Tiny editorial comments:

line 373: (in this case, Hardy-Weinberg equilibrium based on allele frequencies). I don't think this is quite right, as 'Hardy-Weinberg equilibrium' isn't a parameter of the structure model - its an assumption of the method that subpopulations are in HWE.

Line 437: from southwest in rainy season and from northeast in dry season -> "from the southwest in the rainy season and from the northeast in the dry season"

line 438 - the section on seasonal wind direction was just a tiny bit hard to follow - the message is clear once you've read it twice: that in the dry season few flies are around, so the prevailing wind from Metema -> Metekel is unlikely to lead to much migration, while the prevailing wind direction in the rainy season will lead to movement Metekel -> Metema. But I wonder if the authors can revise this to make this message a bit more explicit -

Reviewer's Responses to Questions

**Key Review Criteria Required for Acceptance?**

**Methods**

-Are the objectives of the study clearly articulated with a clear testable hypothesis stated?

-Is the study design appropriate to address the stated objectives?

-Is the population clearly described and appropriate for the hypothesis being tested?

-Is the sample size sufficient to ensure adequate power to address the hypothesis being tested?

-Were correct statistical analysis used to support conclusions?

-Are there concerns about ethical or regulatory requirements being met?

Reviewer #1: The manuscript uses two approaches to analyse the population structure of onchocerciasis vectors in Ethiopean foci. Importantly, the authors include a excellent polytene chromosome analysis. Genome syteny as revealed by polytene chromosome cytogenetics forms the basis of species-level taxonomy in Simuliidae and without itthe literature can become clouded by taxonomically unplaced "genotypes" of unknown affinity. The second prong of this paper, RAD-Seq analysis, has been successfully used in other vector taxa, e.g., mosquitoes. The authors use this newer technique to great effect, demonstrating not only 2 geneetically isolated entitites in the local damnosum s.l. population but also the technique's utility for migtration and re-colonization studies that will be critical for elimination efforts. 

Both approaches were carefullly done and well analysied.

Reviewer #2: General points:

The authors use PCA and STRUCTURE output as the basis for inferring possible long-distance migration between two river basins (previously known to be the basis of Simulium population clustering). The Simulium blackfly is an understudied organism and there is much left to learn about its ecology and genetics - so I was really happy to read this manuscript. This manuscript identifies some interesting signatures of population structure, and potential long-distance migration between two different transmission zones that supports existing genomic work using parasite mitochondrial genome data by one of the groups involved suggesting that blackflies, and Onchocerca parasites, are structured by river basin. As such, it is a useful addition to the field. This is a really interesting dataset, and I felt that there are ample opportunities for more in-depth and informative analyses of these data that would be informative for detecting the extent to which variation in dispersal, and inversion polymorphism frequency, contributes to the signal observed here. The authors are careful to acknowledge the limitations of their study, and couch their claims accordingly. In the absence of further analysis (which I think they could do, and would add substantially to the paper), my main comment is that I feel it could be more grounded in the extensive body of literature that uses genetic data to understand organism movement, as well as the contribution of spatial structure and inversion polymorphism to observed patterns of genetic structure. I think this would add a lot of interesting detail to the background and interpretation of the findings.

E.g.

https://www.biorxiv.org/content/10.1101/2020.03.01.971333v3.full

https://pubmed.ncbi.nlm.nih.gov/36524932/ (and many others like it)

https://onlinelibrary.wiley.com/doi/full/10.1111/mec.15707

https://academic.oup.com/genetics/article/215/1/193/5930485?login=true

https://www.annualreviews.org/doi/abs/10.1146/annurev-ecolsys-110316-022659

I have never performed, analysed any data from, or been involved in, cytogenetic analyses, so I cannot comment on the methods and results for these sections of the paper.

Specific comments below:

Line 86: Do the authors have a citation for this statement?

Line 87/88: There is a lot of theoretical and practical work supporting this in populations generally that could be cited (e.g. Ilkka Hanskki’s work in Glanville Fritillaries, Janet Midega’s 2012 paper on mosquito larval sites and malaria case)

Line 216 (Fig 1). The map is quite grainy - please could a higher-definition map be provided?

Line 254: The authors haven’t pruned their data for linkage disequilibrium (LD). LD can mask signals of population structure, or create spurious signatures of the population subdivision, especially when inversions are concerned. I suggest the authors at least examine the effect of LD pruning on their PCA to see the extent to which pruning for LD changes the PCA result, and if it does, do this for the STRUCTURE analysis too.

Informative blog posts here: https://privefl.github.io/bigsnpr/articles/how-to-PCA.html. http://alimanfoo.github.io/2015/09/28/fast-pca.html

Line 354: I find the PCAs in 3A and B quite difficult to interpret as the plots are overcrowded with lines and ellipses. I appreciate for DAPC this is part of the analysis and interpretation but for ease of interpretation I would include at least the PCA in Fig 3A without lines or ellipses / move the annotated standard PCA to supplementary.

It’s also not clear whether A and B are both DAPC output or not, as ‘standard’ PCA generally doesn’t include lines/ellipses - if Fig 3A is not DAPC, where do the lines and ellipses come from?

It would assist with interpretation of the results if the authors could include PCA results from lower PCs (3,4 and 5,6), as well as the amount of variation in the data explained by each PC.

The lower PC’s could go into supplementary information. This would allow readers to examine the extent to which the PCAs represent variation in the data and also interpret subtle structure often revealed by examining lower PCs.

Line 417: The three groups in the Metekel cluster on the Y axis of Fig 3A look like they could, possibly, be caused by an inversion polymorphism. See, amongst many others:

https://onlinelibrary.wiley.com/doi/10.1111/mec.15428 - hard to tell with these data without them being anchored by chromosome, but sometimes inversions can drive very strong signal in PCA. LD pruning would help resolve whether this is the case.

Line 419: I would be very interested to see how distinct the two clusters are. Examining lower PCs, and quantifying % of variance explained by each PC, as well as quantifying Fst between the different sites and clusters (presented as a heatmap of pairwise between-site Fst), would be informative in this case and easy to achieve with the data and workflows the authors possess. Population structure may vary across the genome, and geographic variation in this can bias analyses such as PCA (inversion polymorphisms are a great example of that - see https://pubmed.ncbi.nlm.nih.gov/30459280/, and the Anopheles literature) to give clustering patterns that reflect inversion frequency as opposed to neutral genetic structure. Certainly when genome-wide data are examined over RAD-Seq / cytogenetic data, previously defined population subdivisions break down in favour of regions of the genome where gene flow is restricted. 

Line 417 - again:

The authors are careful to equivocate between different demographic scenarios underlying the clustering pattern, and appearance samples from both clusters in one of the sampling sites. If the authors wished, they could analyse their data further to identify whether one of their two proposed scenarios (discrete structure with recent migrants, or continuous structure with two distinct, non-interbreeding populations) are more likely. STRUCTURE assigns populations in the presence of spatial autocorrelation in relatedness between geographically close samples. Perhaps blackflies exhibit strong signals of isolation-by-distance, which would account for two populations, sampled far apart, appearing to be from discrete transmission zones/river basins, when actually they are part of a continuously structured population (reflected in the mitochondrial data, which from a relatively short marker and therefore less prone to signatures of isolation-by-distance as will have fewer segregating sites). A test for isolation-by-distance (from kinship, or sitewise Fst) would inform the extent to which spatial structure may be contributing to the clustering pattern in the STRUCTURE output and the extent to which these clusters are actually discrete populations, and then allow for more robust conclusions to be drawn about potential migration between discrete populations - and whether the data are likely to support either of the hypotheses presented in Line 454. The conSTRUCT software package by Gideon Bradburd aims to parse between similar scenarios: https://academic.oup.com/genetics/article/210/1/33/6088031

Perhaps the two (though I actually see four) clusters in the PCA and STRUCTURE are due to geographic variation in inversion polymorphism frequency (e.g. the 3 clusters we see along the y axis of 3A is due to 0/0, 1/0, 1/1 of one inversion, and the single cluster from Metema is 0/0, 1/1 for a different segregating inversion), and individuals with an inversion karyotype more typical for Metekel are present in Metema but at a slightly lower frequency? Examining the effect of LD pruning as suggested above will be informative for this. 

Assigning directionality to these types of data (e.g. “northward migration” is very difficult. There are methods enabling this, for example BayesAss or DisperseNN, if the authors wished to find directionality of migration.

Line 459: Could there have been contamination or sequencing error? 

Line 496: “Limitations of the study” - line left in?

Reviewer #3: the objectives are clear and articulated with the research question

The design is very appropriate and introduce one more approach to identify the species

The population is described clearly

Due to the difficulty in finding suitable stages for identification by cytotaxonomy, this approach remains unresolved, but the authors discuss it very well.

There is not a clear description about ethical requirements, taking into account the authors catched insects landing on human, but they did not explain or described if there are some permission? in Ethiopia are some requirements? Which institution approved this work?

**Results**

-Does the analysis presented match the analysis plan?

-Are the results clearly and completely presented?

-Are the figures (Tables, Images) of sufficient quality for clarity?

Reviewer #1: The results of both approaches are complementary and extremely significant. Both the cytogenomic (polytene chromosome) and RAD-Seq will be valuable for the epidemiology of onchocerciasis in the region, and for control/monitoring efforts. The results are well illustrated, and the figures enhance the reporting of the results. The revealing of a new cytoform (potentially a new species) in the damnosum complex is significant, as is the RAD-Seq demonstration of two non-interbreeding forms. 

I was intrigued by the discovery of a the polymorphic inversion 2L.8, since it parallels the multiple roles of the IIIL-19 inversion in the Simulium vernum complex, which can likewise be fixed autosomal, sex-linked or autosomal polymorphic.

The ambiguity/complexity in the mtDNA bar-coding is also an important result. There is

Reviewer #2: (No Response)

Reviewer #3: Yes, the analysis and the figures are clear

**Conclusions**

-Are the conclusions supported by the data presented?

-Are the limitations of analysis clearly described?

-Do the authors discuss how these data can be helpful to advance our understanding of the topic under study?

-Is public health relevance addressed?

Reviewer #1: The conclusions are well justified by the data. The deomonstration of a new cytoform/species in the damnosum complex and two non-interbreeding forms in the Ethiopean foci is important and timely.

Reviewer #2: (No Response)

Reviewer #3: The authors discuss the biases found and put forward two possible hypotheses that may respond to what they found.

yes, they highlight the importance for public health, especially the possible influence on the onchocerciasis elimination program.

**Editorial and Data Presentation Modifications?**

Reviewer #1: No modifications necessary.

Reviewer #2: (No Response)

Reviewer #3: Minor revision

**Summary and General Comments**

Reviewer #1: In view of the renewed efforts to control and hopefully eliminate onchocerciasis in much of sub-Saharan Africa, this paper is both impportant and timely. It will aid in design of control efforts, not only in the Ethiopean foci but elsewhere by highlighting important vector population considerations. It will also help avoid reliance on fast but dubious bar-coding methods by the paper's demonstration of the complexity of mitochondrial sequence variant distribution within the complex.

Both the bar-coding and polyteene results contribute to the "pure science" understanding of introgression and sharing of polymorphisms.

Reviewer #2: (No Response)

Reviewer #3: there are some gaps in order to understand the hypotheses proposed at the end, for example, it is not known if in the areas between the two subfocus there are possible breeding grounds for the species or if all the stream dry up.

And the other suggestion. 

The discussion should go deeper with more recent articles on the genes that have been used to identify the species and especially the use of CO1 and its ability to discriminate the species, currently there is literature from other regions that address this topic.

PLOS authors have the option to publish the peer review history of their article (what does this mean?). If published, this will include your full peer review and any attached files.

Reviewer #1: No

Reviewer #2: No

Reviewer #3: No

Figure Files:

Data Requirements:

Reproducibility:

References

---

## [Editor Report · Decision Letter 1]

17 Dec 2023

Dear Dr. Wilding,

We are pleased to inform you that your manuscript 'Cytotaxonomic characterization and estimation of migration patterns of onchocerciasis vectors (Simulium damnosum sensu lato) in northwestern Ethiopia based on RADSeq data' has been provisionally accepted for publication in PLOS Neglected Tropical Diseases.

Best regards,

James Cotton

Academic Editor

Nigel Beebe

Section Editor

The authors have done a good job in responding to the reviewers comments, and this is a very nice contribution to the journal. Thanks.

I noticed a couple of tiny points when re-reading the revised manuscript that the authors might want to fix in preparing final files for publication:

line 28 - I think these should be dashes rather than hyphens, as being used to separate clauses.

line 277-281. Any citations for the known inversions mentioend in this section?

line 342 - what is "M = 1"? I think a parameter of Stacks but should be explained somewhere.

Line 366 - 'When adegenet' - the 'when' shouldn't be italicised.

---

## [Editor Report · Acceptance letter]

28 Dec 2023

Dear Dr. Wilding,

We are delighted to inform you that your manuscript, "Cytotaxonomic characterization and estimation of migration patterns of onchocerciasis vectors (Simulium damnosum sensu lato) in northwestern Ethiopia based on RADSeq data," has been formally accepted for publication in PLOS Neglected Tropical Diseases.

Best regards,

Shaden Kamhawi

co-Editor-in-Chief

Paul Brindley

co-Editor-in-Chief
